# The Potential of *Platanus orientalis* L. Bark for High-Grade Resource Utilization

**Hanyin Li** [1], **Yunming Zou** [1], **Jingyi Liang** [1,2], **Zijie Zhao** [1], **Na Zhou** [1], **Yan Gao** [1], **Ruohan Yan** [1], **Qiongqiong Zhou** [3,*] **and Cheng Li** [4,*]

1   College of Forestry, Henan Agricultural University, Zhengzhou 450002, China; lihanyin@henau.edu.cn (H.L.); zouym129@163.com (Y.Z.); l374811491@163.com (J.L.); hnzhaozijie@163.com (Z.Z.); 18539954121@163.com (N.Z.); yy15839337679@163.com (Y.G.); yanruohan0375@163.com (R.Y.)
2   College of Landscape Architecture and Art, Henan Agricultural University, Zhengzhou 450002, China
3   College of Horticulture, Henan Agricultural University, Zhengzhou 450002, China
4   Forest Biomass High-Value Engineering Technology Research Center of Henan, Henan Agricultural University, Zhengzhou 450002, China
*   Correspondence: zqxy1223@henau.edu.cn (Q.Z.); lichengzzm@163.com (C.L.)

**Abstract:** Forest wood biomass can be used as a renewable resource for the sustainable production of fuels and chemicals. In this study, the methanol, methanol/ethanol, and ethanol/benzene solvent extracts of *Platanus orientalis* L. bark were analyzed using FTIR, $^I$H NMR, $^{13}$C NMR, 2D-HSQC NMR, GC-MS, and TOF-LC-MS. The results revealed that the bark of *Planus orientalis* contained a wide variety of chemical compounds, such as 30-triacontanol, 1-Hexanol, hexadecanoic acid, methyl ester, 2-ethyl-, γ-Sitosterol, and 3,4,5-tri methoxy-Phenol. In addition, the fast pyrolysis of *P. orientalis* L. bark (POL-B) with nano-catalysts ($Co_3O_4$, $Fe_2O_3$, and $Co_3O_4/Fe_2O_3$) was investigated using pyrolysis/gas chromatography/mass spectrometry (Py-GC/MS) and a thermogravimetric analyzer coupled with an FTIR spectrophotometer (TG-FTIR). The TG results revealed that the nano-catalysts significantly affected the pyrolysis of *P. orientalis* bark. The nano-$Fe_2O_3$ catalyst was shown to increase acid and ketone compound production during the catalytic pyrolysis of cellulose. According to the Py-GC-MS results, the pyrolytic products contained several value-added chemicals and high-quality bio-oil. The nano-catalysts promoted the production of aromatics, phenols, ketones, olefins, furans and alkane compounds. These natural-product active molecules and bio-oil, as high-grade raw materials, could be used in many industrial and agricultural fields for the production of wetting agents, stabilizers, plasticizers and resins. In addition, a number of active molecules could be used as drugs and biomedical active ingredients for anti-cancer and anti-inflammatory purposes.

**Keywords:** forest wood biomass; fuel; nano-catalysts; pyrolysis; forest waste

## 1. Introduction

The continuous growth of the global population has triggered a global energy crisis [1]. The problems of environmental pollution [2], ecological damage, and climate change [3,4] have become increasingly serious [5]. Biomass, for the sustainable production of chemicals and fuels [6], is extremely abundant in nature [7]. It typically includes crops, crop waste straw, industrial forest waste, and aquatic plants [8]. Usually, biomass is burned and thus used to obtain energy, which not only causes considerable waste but also leads to environmental pollution. Therefore, converting biomass into chemical fuels or high-value chemicals using chemical conversion technologies (such as pyrolysis and liquefaction) has attracted attention [9].

Transforming biomass into value-added products can be realized through the conversion of platform molecules [1,10]. The main problem for the transformation process is the lack of efficient catalytic materials because biomass platform molecules often have very high molecular weight and oxygen content. The catalytic material required for biomass

conversion differs from that of the traditional petrochemical catalytic material. Currently, the main platform compound conversion pathway uses enzyme catalysis or a homogeneous catalytic system to obtain the target product [11–13]. However, the enzyme catalytic system has high input costs and equipment requirements and suffers from issues such as the need for waste-liquid treatment, which often fails to achieve the required industry standard for biomass conversion. The homogeneous reaction system often needs the addition of soluble acid, alkali, and salt as catalytic materials, which not only corrodes the equipment but is also difficult to separate from the reaction system. Researchers have developed many heterogeneous catalytic materials for catalyzing the conversion reactions of biomass platform compounds [14]. However, most of these suffer from problems such as low selectivity and activity [15] and poor stability, resulting in difficulties in applying these catalytic materials in large-scale industrial operations. In addition, the biomass platform molecules generally have multiple functional groups and are highly active, which makes them very prone to side reactions. Therefore, developing catalytic materials with a high activity and selectivity would be very promising and interesting [16].

Fast pyrolysis is a good way to utilize biomass resources. The economic benefits of fast pyrolysis are 2–3 times those of biomass gasification and liquefaction [17–19]. The bio-oil obtained from the fast pyrolysis of lignocellulosic biomass has drawbacks such as high acidity and viscosity, a low heating value, and unsuitable storage capability, which restricts its utilization as a raw material for chemical products or as transportation fuel [20–22]. In the studies of Ge et al., *Pinus armandii* Franch was pyrolyzed and extracted to produce natural active molecules, bio-oil, and products. They found that the natural active molecules and bio-oil of Pinus armandii Franch could be used as raw materials for drugs and biomedical active ingredients. The catalytic conversion of pyrolysis steam can effectively improve bio-oil quality [17,23,24]. To date, catalytic deoxygenation has been extensively investigated for more than three decades [25,26]. Zeolite-based molecular sieves are widely used for the deep catalytic deoxidation of bio-oil or pyrolysis vapor [27,28]. When such catalysts are used for the catalytic pyrolysis of biomass, they exhibit high-efficiency deoxidation characteristics [29]. For example, the HZSM-5 molecular sieve can effectively increase the content of aromatic hydrocarbons in bio-oil [30]. Among the applications of fast pyrolysis, microwave-assisted catalytic liquefaction and high-pressure liquefaction technologies have made considerable progress in recent years [24,31]. Several studies have shown that lignin depolymerization can efficiently be used to obtain aromatic hydrocarbon compounds [31,32]. Nanoparticles have been used as catalysts in biomass pyrolysis to upgrade the fuel properties of bio-oil and enhance the formation of valuable chemicals [33,34]. Lu et al. [34] reported that the catalytic effect of nano-CaO significantly reduced the content of phenols and anhydro sugars, eliminated acidic substances, and increased the formation of cyclopentanones, hydrocarbons, and several light compounds. In addition, catalysis via nano-$Fe_2O_3$ resulted in the formation of various hydrocarbons. However, nano-ZnO slightly altered the pyrolytic products. Khelfa et al. reported that $Fe_2O_3$ could break down the tar produced and improve the partial oxidation of phenols during the thermal degradation of the biomass [35]. The study of Li et al. showed that the $Fe_2O_3$ and NiO catalysts enhanced the pyrolysis process by accelerating the precipitation of gaseous products [31].

Bark is a hugely abundant lignocellulosic biomass type with a variety of bioactivities in its extracts, including cardiovascular benefits, antioxidants, and anti-diabetic effects [36,37]. However, the potential value of the bark extracts as biomedicine or fuel depends on the chemical and physical characteristics of the biomass, which is affected by the species type, geographic location, and local weather. Therefore, conducting a full-component study of biomass is particularly important to enhance the added value of the residue biomass and maximize its economic and ecological benefits [38].

This study aimed to understand the detailed pyrolytic products and extractives of *Platanus orientalis* L. wood bark. TG and TG-FTIR were used to analyze the degradation properties of *P. orientalis* bark. *P. orientalis* bark (POL-B) has good potential as a renewable

resource for the sustainable production of fuels and chemicals; hence, the fundamental chemistry of its extractives will provide valuable insights for the development of natural products for biomedical applications [39].

## 2. Materials and Methods

### 2.1. Experimental Section

Materials: *P. orientalis* L. bark (POL-B) was collected from the campus of Beijing Forestry University, Beijing, China. Anhydrous sodium sulfate, benzene, methanol, and ethanol were purchased from Xilong Science Reagent Co., Ltd. (Tianjin, China). All reagents were of chromatographic grade. All nano-catalysts were purchased from Tianjin Kemiou Chemical Reagent Co., Ltd. (Tianjin, China). The parameters of the nano-catalysts were as follows: nano-$Fe_2O_3$ (MACKLIN, 99.5%, $\alpha$-$Fe_2O_3$, 30 nm, spherical), nano-$Fe_3O_4$ (MACKLIN, 99.5%, 60–120 nm) and nano-$Co_3O_4$ (MACKLIN, 99.9%, 80 nm). The compounding ratios of catalysts and POL-B are listed in Table 1.

**Table 1.** The compounding ratio of catalysts and POL-B.

| Samples | Barks (g) | $Co_3O_4$ (g) | $Fe_2O_3$ (g) | $Co_3O_4$ + $Fe_2O_3$ (g) |
|---|---|---|---|---|
| POL-B | 20.00 | | | |
| POL-B- $Co_3O_4$ | 20.00 | 0.20 | | |
| POL-B- $Fe_2O_3$ | 20.00 | | 0.20 | |
| POL-B- $Co_3O_4/Fe_2O_3$ | 20.00 | | | 0.10 + 0.10 |

Extraction: 40 g POL bark was crushed into 100 mesh, filled into a cotton bag, and signed. The extracts from POL bark were obtained using the Soxhlet extraction method [40,41]. The conditions were as follows: 300 mL methanol, methanol/ethanol, and ethanol/benzene (1:1) as solvents, respectively [42]; the extraction time was 6 h; the temperature was 60 °C [43]. Subsequently, the solvents were removed with the rotary method, and then, the extractives were obtained and stored at −5 °C [44].

### 2.2. Characterizations

The extractives of the bark of POL were characterized using FTIR and NMR. NMR spectra were performed on a Bruker AVIII 400 MHz spectrometer. For $^{13}C$ spectrogram acquisition (procedure: $C_{13}IG$), 10 mg of the extractive samples were weighed and dissolved in DMSO-d6, and 25 µL of chromium (III) acetylacetonate (concentration = 0.01 M) was added. For the acquisition of the two-dimensional hydrocarbon direct correlation spectrum (2D-HSQC NMR), all samples were crushed to a 100 mesh size and then underwent vacuum free-drying at −60 °C for 24 h. A weight of 5 mg of the sample was weighed. The thermogravimetric curve was measured from room temperature to 950 °C using a thermogravimetric analyzer (TGA-Q50, TA Instruments, New Castle, DE, USA) under $N_2$ conditions with heating rates of 20 °C/min, 60 °C/min, and 90 °C/min, respectively [45]. GC/MS analysis conditions were as follows: the chromatographic column was a quartz capillary column (30 mm × 0.25 mm × 0.25 µm). The mass spectrum program scanned the mass range from 30 to 600 amu, and a wiley7n.1 standard spectral method and computer retrieval qualitative method were used for analysis [46,47]. LC-RP-QTOF-MS analysis of all extractive samples was performed using a Bruke 1290 HPLC and a 6550 QTOF detector. The following LC and MS parameters were provided by Bruke Co. (Beijing, China). LC: The column was Agilent Eclipse Plus C18 (2.1 mm × 100 mm, 1.8 µm). Elution was performed at a flow rate of 0.30 mL/min for 5 min at 40 °C. MS: Drying gas temperature: 200 °C/325 °C; sheath gas temperature: 350 °C; scanning quality range program: 50–1200 $m/z$. A TGA Q500 (TA Instrument, USA) connected with a Fourier-transform infrared spectrometer (Nicolet 6700, Nicolet Instrument Corporation, Madison, WI, USA) was used to detect the degradation properties of all samples. TG-FTIR analysis of poplar bark (POL-B), bark containing $Co_3O_4$ (POL-B-$Co_3O_4$), bark containing $Fe_2O_3$

(POL-B-Fe$_2$O$_3$), and bark containing Co$_3$O$_4$/Fe$_2$O$_3$ (POL-B-Co$_3$O$_4$/Fe$_2$O$_3$) was performed. A weight of 6 mg of the sample was weighed, the heating rate was 60 °C/min under a nitrogen environment, and the temperature range was from 30 °C to 950 °C. The IR-spectra wavenumber range was from 4000 to 400 cm$^{-1}$ with a resolution of 4 cm$^{-1}$, and 3D FTIR spectrograms were obtained. The rapid pyrolysis of POB bark was performed using pyrolysis (Curie pyrolyzer JHP-22, Analytical Industry Co., Ltd., Tokyo, Japan) followed by gas chromatography (Agilent 7200B Q-TOF GC/MS, Agilent Technologies, Inc., Santa Clara, CA, USA)/mass spectrometry (Agilent 7200B Q-TOF GC/MS, Agilent Technologies, Inc., USA). The schematic diagram of useful products that can be obtained from POL bark extractives and pyrolytic products is shown in Figure 1.

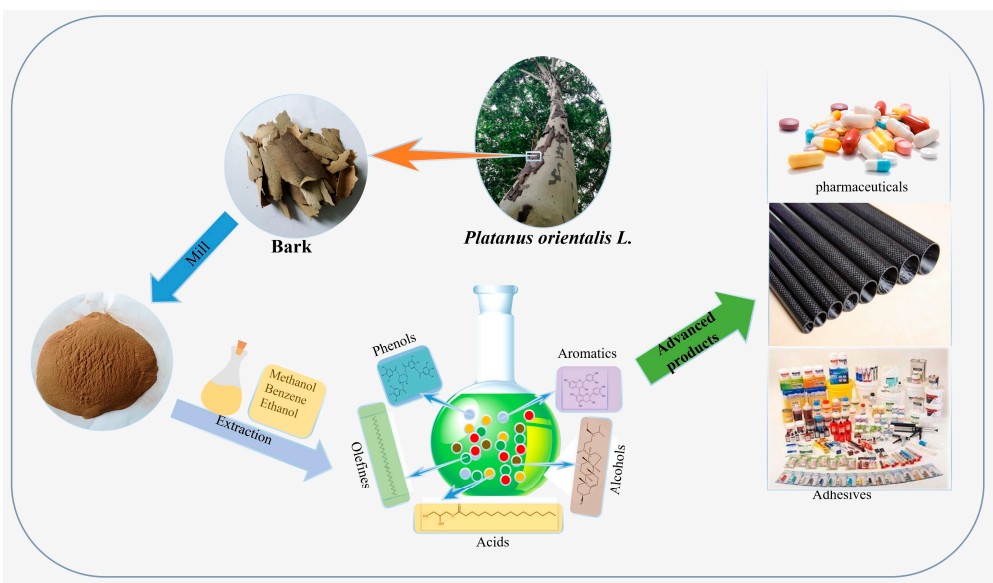

**Figure 1.** Schematic diagram for converting POL bark extractives into advanced products.

## 3. Results and Discussion

### 3.1. Characterization of POL Bark Extracts

3.1.1. Characterization of POL Bark Extracts using GC/MS

Figures S1–S3 and Tables 2–4 represent a typical GC/MS total ion chromatogram for *P. orientalis* bark extractive. These compounds were classified based on their functional groups, such as acids, alcohols, aldehydes, esters, amides, phenols, saccharides, ketones, aromatics, olefins, and ethers. Tables 2–4 show that different types of extractants can extract various compounds, while the main types of compounds are almost similar. From Table 2, the main chemical composition of the methanol extractives of *P. orientalis* bark were acids (9.708%), alcohols (34.421%), amides (2.391%), phenolics (1.600%), aromatics (18.753%), and olefines (33.125%). For Table 3, the pivotal chemical composition of the ethanol/benzene extractives from *P. orientalis* bark were acids (6.235%), alcohols (55.821%), esters (11.585%), saccharides (0.939%), amides (18.141%), ketones (3.787%), olefines (1.471%), and ethers (2.201%). The main chemical composition of the methanol/ethanol extractives of *P. orientalis* bark (Table 4) were acids (5.538%), aldehydes (0.796%), alcohols (23.567%), esters (17.133%), saccharides (1.691%), amides (2.150%), phenolics (7.048%), aromatics (40.609%), and olefines (4.318%).

As stated above, different solvents resulting in various extractive compounds indicate that using multiple solvents could improve extraction yield. The extractive findings revealed that the *P. orientalis* bark contains a variety of chemical constituents with a wide range of potential applications. For instance, the extractives of *P. orientalis* bark contained several acidic compounds, such as oleic acid, hexadecanoic acid, 2-methyl-, n-hexadecanoic acid, and vanillic acid, which can be used as raw materials for soaps, candles, lubricants,

softeners, and synthetic detergents. The alcoholic compounds, such as 30-triacontanediol, 1-hexanol, and 2-ethyl-, can be used for medicines that stop bleeding or for dehumidification and detoxification. In addition, methyl stearate, classified under ester compounds, can be used as a surfactant and lubricant. Hexadecanoic acid and methyl ester are only two examples of ester compounds that may be utilized as building blocks in emulsifiers, wetting agents, stabilizers, and plasticizers. Resins, synthetic fibers, refined oil, plastics, medicines, and insecticides may all use phenolic compounds as raw materials. Substitution reactions allow aromatic compounds to synthesize more complex molecules than simpler ones. For the synthesis of polyolefins and synthetic rubbers, organic synthesis is significantly based on the use of olefine compounds as fundamental building blocks. As a result, the chemical components of the *P. orientalis* bark offer potential new materials for application in various industrial and agricultural sectors, as well as in the manufacture of bio-oil, pharmaceuticals, and medical treatments. Sitosterol compounds were found in the extracts of ethanol/benzene and methanol/ethanol, which has excellent antifungal activity [48], particularly against *Aspergillus niger*, *Cladosporium cladosporioides* and *Phytophthora* sp. *fungi* [49,50]. Large amounts of phenols and aromatic compounds were found in the methanol/ethanol extracts (Table 4), indicating that the bark of *P. orientalis* contained a large amount of tannin, which can be used as a raw material for resins, adhesives, metal-iron absorption, and bio-medicines.

**Table 2.** Detailed composition of the extractives of POL bark by methanol.

| Compound | Concentration (wt.%) |
|---|---|
| **Acid** | 9.708 |
| 9,12-Octadecadienoic | 1.844 |
| 9-Hexadecenoic, acid | 3.549 |
| n-Hexadecanoic, acid | 2.176 |
| Oleic acid | 2.139 |
| **Alcohols** | 34.421 |
| 1-Hexanol, 2-ethyl- | 1.982 |
| 6-Isopropenyl-4,8a-dimethyl-1,2,3,5,6,7,8,8a-octahydro-naphthalen-2-ol | 1.196 |
| 1,30-Triacontanediol | 28.331 |
| Cryptomeridiol, | 1.606 |
| 7,8-Epoxylanostan-11-ol-3-acetoxy- | 1.306 |
| **Amides** | 2.391 |
| Formamide, N,N-diethyl- | 2.391 |
| **Phenol** | 1.600 |
| (E)-4-(3-Hydroxyprop-1-en-1-yl)-2-methoxyphenol, | 0.743 |
| Phenol, 3,4,5-trimethoxy- | 0.857 |
| **Aromatics** | 18.753 |
| 5H-Cyclopropa[3,4]benz[1,2-e]azulen-5-one, 9,9a-bis(acetyloxy)-3-[(acetyloxy)methyl]-1,1a,1b,2,3,4,4a,7a,7b,8,9,9a-dodecahydro-2,3,4a,7b-tetrahydroxy-1,1,6,8-tetramethyl-, | 0.687 |
| 1H-Cyclopropa[3,4]benz[1,2-e]azulene-4a,5,7b,9,9a(1aH)-pentol, 3-[(acetyloxy)methyl]-1b,4,5,7a,8,9-hexahydro-1,1,6,8-tetramethyl-, | 7.145 |
| 4H-Cyclopropa[5′,6′]benz[1′,2′:7,8]azuleno[5,6]oxiren-4-one, 8,8a-bis(acetyloxy)-2a-[(acetyloxy)methyl]-1,1a,1b,1c,2a,3,3a,6a,6b,7,8,8a-dodecahydro-6b-hydroxy-3a-methoxy-1,1,5,7-tetramethyl-, | 9.812 |
| Benzenemethanol, 3,4,5-trimethoxy- | 1.109 |
| **Olefine** | 33.125 |
| 17-Pentatriacontene | 33.125 |

**Table 3.** Detailed composition of the extractives of POL bark by ethanol/benzene.

| Compound | Concentration (wt.%) |
|---|---|
| **Acid** | 6.235 |
| Hexadecanoic acid,2-methyl- | 4.646 |
| elaidate | 1.589 |
| **Alcohols** | 55.821 |
| β-Sitosterol | 7.389 |
| γ-Sitosterol | 0.91 |
| 1-Hexanol,2-ethyl- | 15.582 |

**Table 3.** *Cont.*

| Compound | Concentration (wt.%) |
|---|---|
| 1H-Cyclopropa[3,4]benz[1,2-e]azulene-5,7b,9,9a-tetrol, 1a,1b,4,4a,5,7a,8,9-octahydro-3-(hydroxymethyl)-1,1,6,8-tetramethyl-,5,9,9a-triacetate, | 2.601 |
| 1H-Cyclopropa[3,4]benz[1,2-e]azulene-4a,5,7b,9,9a(1aH)-pentol, 1b,4,5,7a,8,9-hexahydro-3-(hydroxymethyl)-1,1,6,8-tetramethyl-, | 19.811 |
| 1,22-Docosanediol | 9.528 |
| **Esters** | 11.585 |
| Heptadecanoic acid, 15-methyl-, methyl ester | 3.161 |
| i-Propyl 14-methyl-pentadecanoate | 2.432 |
| 1b,4,5,7a,8,9-hexahydro-3-(hydroxymethyl)-1,1,6,8-tetramethyl-, 9,9a-diacetate, | 2.322 |
| 11-Octadecenoic acid, methyl ester | 2.803 |
| 1,2-Benzenedicarboxylic acid, bis(2-methylpropyl) ester | 0.867 |
| **Saccharides** | 0.939 |
| 1-Hydroxy-2-(2,3,4,6-tetra-O-acetyl-β-d-glucopyranosyl)-9H-xanthene-3,6,7-triyl triacetate | 0.939 |
| **Amides** | 18.141 |
| N,N-diethyl-formamide | 18.141 |
| **Ketones** | **3.787** |
| 17.β.-Acetoxy-1′,1′-dicarboethoxy-1.β.,2.β.-dihydrocycloprop[1,2]-5α-androst-1-en-3-one | 3.203 |
| 1,9-Dioxacyclohexadeca-4,13-diene-2-10-dione,7,8,15,16-tetramethyl- | 0.584 |
| **Olefine** | 1.471 |
| 2,5,5,8a-Tetramethyl-4-methylene-6,7,8,8a-tetrahydro-4H,5H-chromen-4a-yl hydroperoxide | 0.547 |
| 17-Pentatriacontene | 0.924 |
| **Ethers** | 2.021 |
| 2,5-dimethoxyphenylethylsulfide | 0.734 |
| (Z)-18-Octadec-9-enolide | 1.287 |

**Table 4.** Detailed composition of the extractives of POL bark by methanol/ethanol.

| Compound | Concentration (wt.%) |
|---|---|
| **Acids** | 5.538 |
| n-Hexadecanoic acid | 4.700 |
| Vanillic acid | 0.838 |
| **Aldehydes** | 0.796 |
| 3,5-Dimethoxy-4-hydroxycinnamaldehyde | 0.796 |
| **Alcohols** | 23.567 |
| γ-Sitosterol | 19.319 |
| 2-ethyl-1-Hexanol | 1.848 |
| 1-Heptatriacotanol | 0.968 |
| 7,8-Epoxylanostan-11-ol,3-acetoxy- | 1.432 |
| **Esters** | 17.133 |
| 9-Octadecenoic acid, methyl ester, (E)- | 4.100 |
| 2-Oxo-1-(3-oxo-butyl)-cyclohexanecarboxylic acid, ethyl ester | 0.781 |
| 7-Methyl-Z-tetradecen-1-ol acetate | 0.847 |
| Hexadecanoic acid, methyl ester | 5.827 |
| Methyl stearate | 5.578 |
| **Saccharides** | 1.691 |
| d-Mannose | 0.814 |
| β-D-Glucopyranose,.β.-D-Glucopyranose,4-O-.β.-D-galactopyranosyl- | 0.877 |
| **Amides** | 2.150 |
| N,N-diethyl-Formamide | 2.150 |
| **Phenols** | 7.048 |
| 3,4,5-trimethoxy-Phenol | 1.817 |
| **Aromatics** | 40.609 |
| 1H-Cyclopropa[3,4]benz[1,2-e]azulene-4a,5,7b,9,9a(1aH)-pentol,1b,4,5,7a,8,9-hexahydro-3-(hydroxymethyl)-1,1,6,8-tetramethyl-,9,9a-diacetate, | 5.231 |
| 4H-Cyclopropa[5′,6′]benz[1′,2′:7,8]azuleno [5,6-b]oxiren-4-one,8-(acetyloxy)-1,1a,1b,1c,2a,3,3a,6a,6b,7,8,8a-dodecahydro-3a,6b,8a-trihydroxy-2a-(hydroxymethyl)-1,1,5,7-tetramethyl-, | 28.084 |
| 5H-Cyclopropa[3,4]benz[1,2-e]azulen-5-one,9,9a-bis(acetyloxy)-3-[(acetyloxy)methyl]-2-chloro-1,1a,1b,2,3,4,4a,7a,7b,8,9,9a-dodecahydro-3,4a,7b-trihydroxy-1,1,6,8-tetramethyl-, | 5.429 |
| γ-Sitostenone, | 1.865 |
| **Olefines** | 4.318 |
| 3-Buten-2-ol,2-methyl-4-(1,3,3-trimethyl-7-oxabicyclo[4.1.0]hept-2-yl)- | 1.126 |
| 3-Buten-2-ol,3-Buten-2-ol,2-methyl-4-(1,3,3-trimethyl-7-oxabicyclo[4.1.0]hept-2-yl)- | 3.192 |
| **Unidentified** | 2.381 |

3.1.2. FTIR Analysis of POL Bark Extracts

The examination and determination of functional groups of components in biomass materials are enabled using Fourier-transform infrared spectroscopy (FTIR). To characterize the chemical structure and groups present in *P. orientalis* bark, FTIR spectra were collected from three different extractives: methanol, ethanol/benzene, and methanol/ethanol. Significant peaks in the spectra have been confirmed to be functional groups. Figure 2 shows the aromatic and aliphatic stretching vibrations responsible for the wide absorption band seen in the 3402–3445 cm$^{-1}$ [51,52]. Peaks at 2868–2900 cm$^{-1}$ are associated with the C-H stretching vibration of alkanes or the anti-symmetric stretching vibration of -CH$_2$ and the Fermi resonance of -CH, while the band at 2928–2978 cm$^{-1}$ is associated with the C-H stretching vibration of -CH$_3$ or -CH$_2$ and -CH- groups [53]. The presence of phenolic chemicals in the *P. orientalis* extract was confirmed by the existence of absorption peaks at 1634, 1650, and 1642 cm$^{-1}$, which corresponds to aromatic ring vibrations [54]. The phenolic-OH stretch vibration and aliphatic CH deformation in methyl groups are responsible for the peak at 1374–1385 cm$^{-1}$ [55]. Lignin is often absorbed this way. A C-O-C asymmetric stretch vibration is associated with the 1271 cm$^{-1}$ absorption. An aromatic CH in-plane bending vibration and the C-O stretching vibrations of aldehydes are responsible for the bands seen at 1165 and 1045 cm$^{-1}$, respectively [56,57]. Additionally, the absorption peak at 879 and 878 cm$^{-1}$ is associated with the powerful C-H vibration [58,59].

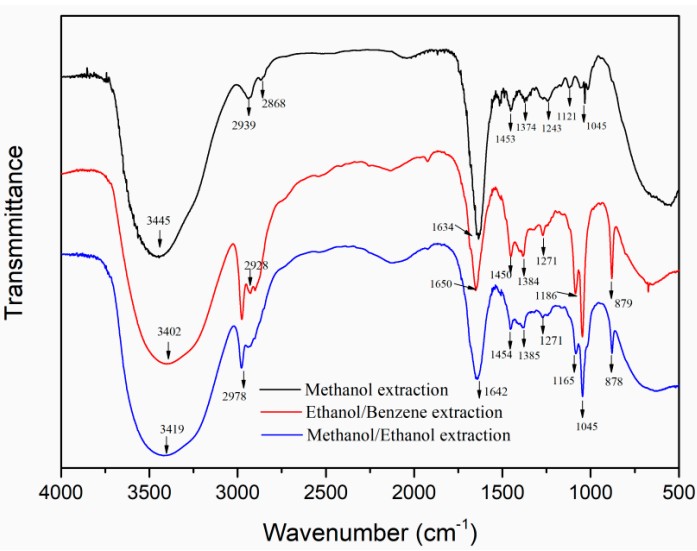

**Figure 2.** FTIR spectra of the methanol, ethanol/benzene, and methanol/ethanol extractives of *Platanus orientalis* Linn bark.

Comparing the FTIR spectra of the different solution extractives of *P. orientalis* bark shows that the ethanol/benzene extractives of *P. orientalis* bark have more substantial absorption peaks than methanol and methanol/ethanol extractives. This indicates that the solution of ethanol/benzene can extract more compounds than methanol and methanol/ethanol solutions. Based on the results of this study, it seems that *P. orientalis* bark appeared rich in a wide range of chemical compounds. These included aromatics, aliphatics, alkanes, aldehydes, ketones, carboxylic acids, and esters.

3.1.3. Characterization of POL Bark Extracts using [I]H NMR, [13]C NMR, and 2D-HSQC NMR

To obtain more detailed information on the chemical structure of the extractions of *P. orientalis* bark, further characterization was conducted using [I]H NMR, [13]C NMR, and 2D-HSQC NMR. The spectra of all extracts are presented in Figures 3 and S1. Based on the previous literature, most chemical shifts could be assigned to the corresponding functional groups.

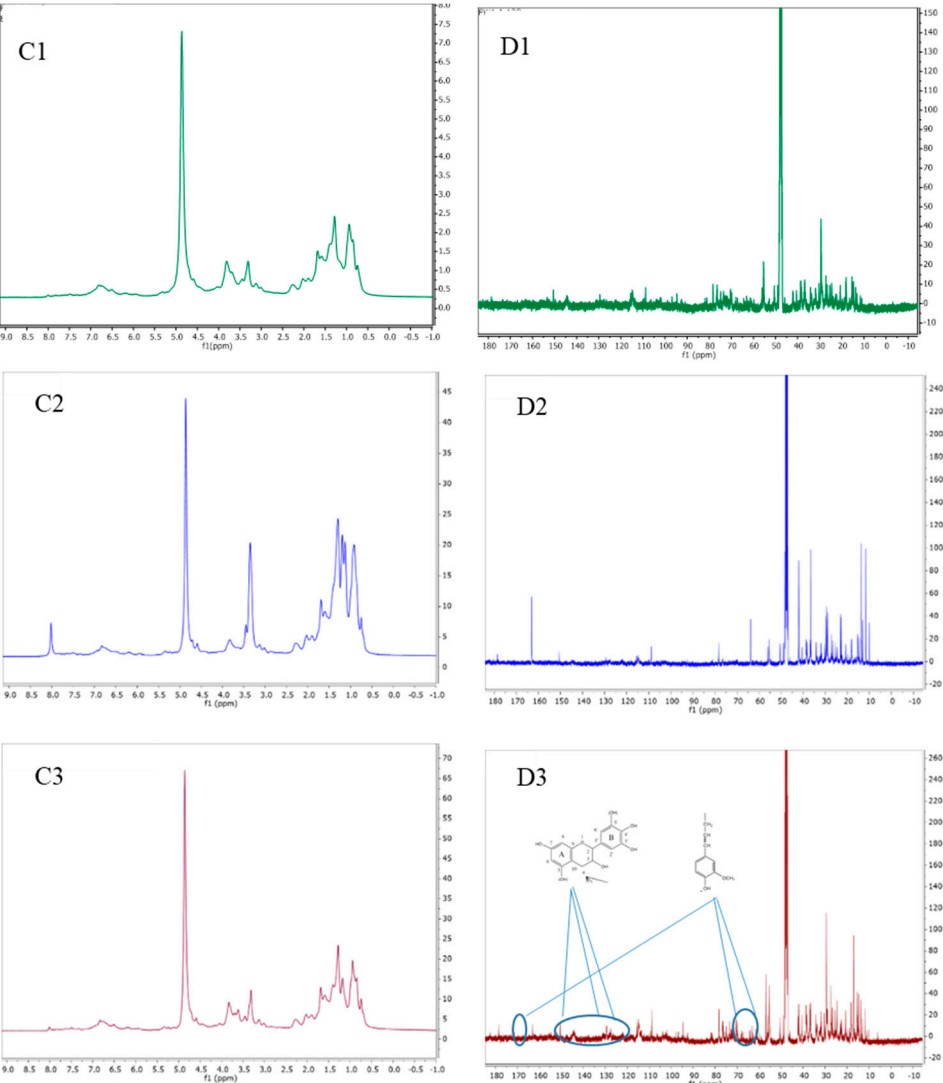

**Figure 3.** $^1$H, $^{13}$C-NMR spectra of the extract of *Platanus orientalis* Linn bark with different solvents: methanol (**C1,D1**), ethanol/benzene (**C2,D2**), and methanol/ethanol (**C3,D3**).

According to $^1$H-NMR spectroscopy, the chemical shifts of protons in saturated alkane compounds fell within the range of δ0.2–δ1.5. The two saturated alkanes that behave like chains had their maxima at δ = 0.8 and δ = 1.3. Proton chemical shifts on carbon atoms directly linked to a halogen fell within 2.0–4.4 ppm. Chemical shifts of 2.0–2.6 ppm were seen for carbonyl -H in carboxylic acid compounds and -H of the hydrogen atom in the ether molecule. Additionally, when conjugated to an aryl group, the chemical shifts of olefinic compounds increased from 4.5 ppm to 5.0 ppm, with increased intensity. This is due to the anisotropic nature of C=C and the sp$^2$ hybridization of carbon in olefins. In addition, carbon protons in alkynes showed chemical shifts of 1.7–3.5 ppm, whereas that of aromatic compounds was 6.1–8.1 ppm. The range of the δ proton concentration in the phenolic hydroxyl group was 4.0–8.1 ppm. The peak intensity of the ethanol/benzene extract (C2) was significantly higher than that of the methanol (C1) and methanol/ethanol (C3) extracts, indicating the effectiveness of the ethanol/benzene extract.

The $^{13}$C-NMR spectrum of *P. orientalis* bark extracts through methanol, ethanol/benzene, and methanol/ethanol analysis indicated that *P. orientalis* bark contains carboxylic acid groups, aromatics, phenols, alcohols, and esters. The chemical shifts at 183 ppm can be attributed to the carbonyl groups in the carboxylic acid groups and the structure of esters. The 178 ppm chemical shift in Figure 3D2,D3 may be explained by the

catechin or epicatechin gallate compounds in tannin [60]. C5 and C7 linked to the phenolic-OH groups on the flavonoid A-ring were the cause of the 150 ppm chemical changes. For the flavonoid B-ring and A-ring, the corresponding chemical shifts were located at 129, 115–116, and 108–114 ppm, respectively, for C1′, C2′, and C5′, and C4–C8; C in the -O-4 structure accounted for the 61–63 ppm shift; the 70–74 ppm shift was characteristic of lignin's guaiacyl and syringyl unit structures; and the 69.8 ppm shift was due to CH-4 in the xylose nonreducing end unit [60,61]. Tannins, phenols, esters, and carboxylic acids were all present in the *P. orientalis* bark extracts, as shown by the $^{13}$C-NMR spectra. Catechin and epicatechin gallate, both flavonoids, have been found to exhibit diverse biological functions in safe amounts in living animals. Cancer prevention through nutrition is a hotly debated topic. There is strong evidence from animal research, human clinical trials, and epidemiological analyses indicating that flavonoids are vital in cancer chemoprevention and chemotherapy [62]. Antiviral, anticancer, antioxidant, antihistaminic, anti-inflammatory, and hepatoprotective characteristics are only some of the pharmacological reactions attributed to flavonoid molecules [63]. Extracted ethanol/benzene HSQC 2D NMR spectra are shown in Figure S1. After extracting different condensed tannins, the broad chemical shifts split into four signals. Aromatic carbons were associated with the signals at H: 6.5–7/C: 110–125 ppm. Aliphatic carbons were responsible for the signal at H: 1–2.5/C: 10–40 ppm. Aliphatic carbons from saccharides were responsible for the signal at H: 3–4/C: 55–65 ppm. A higher-resolution spectrum is required to determine whether the signal at 6.6 ppm was caused by C4-C6 interflavonoid coupling. The extraction with methanol and ethanol had similar results, as seen in Figure S4E1–E3. The presence of tannins was confirmed by signals at H: 4–5/C: 108 ppm, which align with the aromatic carbons in tannin [64,65].

### 3.1.4. Characterization of POL Bark Extracts using TOF-LC-MS

To obtain more detailed information on the chemical structure of the extractions of *P. orientalis* bark, further characterization was conducted using TOF-LC-MS. According to the TOF-LC-MS result of *P. orientalis* bark (Table S1), more than 200 different chemical compounds were identified in 409 peaks of *P. orientalis* bark. Thus, it was indicated that TOF-LC-MS is more suitable for analytical evaluation than FTIR, GC-MS, and NMR, and this method can be used to structurally elucidate as many compounds as possible to obtain more information on general structures and compound types.

The results of the TOF-LC-MS analysis (Table S1) revealed that the high-content extract compounds include β-Apopicropodophyllin, Eurycarpin A, Bufotenidine, Anhydronotoptoloxide, Curcumin, Didrovaltratum, Glycyrrhisoflavone, Kraussianone 2, Betaine, Erypoegin E, Phthalic anhydride, Kraussianone 3, N,N-Dimethyltryptamine N-oxide, Bavachin, Erypoegin A, Physcion, Noradrenaline, Echinopsine, Hispaglabridin B, Lactucopicrin, 5,6-Dimethoxy-7-hydroxycoumarin, Eleutherol, Eryvarin F, Cibarian, Chamanetin, Prenyl caffeate, Methylanthranilate, Celereoin, Isopsoralidin, Epicatechin, Macrophyllic acid, Erycibelline, Diisocapryl phthalate, Aucubigenin, 2,4-Dimethoxybenzaldehyde, Kushenol L, Ditertbutyl phthalate, Fraxetin, Dunnisinin, Procyanidin C, Erosone, Hypaphorine, Glisoflavanone, Phenethylamine, Holadysine, Pfaffic acid, Mammeigin, Gardendiol, Adenine, Theaspirone, Mallotochromene, Lucidin, Thespesone, Desmosflavone, and Glycyrin. Among these compounds, β-Apopicropodophyllin can potentially be used as an agent for cancer treatment. Kim et al. reported that beta-Apopicropodophyllin (APP) is responsible for the cell death of non-small cell lung cancer (NSCLC) cell lines [66]. Eurycarpin A is an effective component of licorice, which can treat coughs and improve digestive disorders [67]. Bufotenidine has an anti-acetylcholine action and can promote histamine release. Phthalic anhydride is helpful in the manufacturing of chemicals, including plasticizers. Lucidin can be used as an indicator for chemical analysis and biological stains. For example, it can be a substrate to detect firefly luciferase activity. Thespesone has the potential as a raw material to treat diabetes mellitus and cancer [68]. Adenine is a component of nucleic acids involved in the synthesis of genetic material. It can promote leukocyte proliferation, increase the number of white blood cells, and be used to prevent leukopenia caused by various

causes, especially leukopenia caused by tumor chemotherapy, as well as acute neutropenia. Physcion is a highly active plant-derived fungicide extracted from the active ingredients of natural plant rhubarb, which is very effective for controlling powdery mildew, downy mildew, gray mold, anthracnose, etc. Due to its low toxicity and friendliness to humans and animals, it is safe to apply as a pesticide agent in the production of green organic vegetables. According to Jean Louis et al., terpene compounds have several roles in the tree, such as resistance to diseases and microbial attacks and the creation of odor (nerolidol, farnesol, and cedrol). They found that triterpenes like lupeol, botulin, and betulinic acid can be extracted from bark to achieve anticancer properties [48,49].

According to the findings, *P. orientalis* bark contains various chemical constituents, including aromatics, alkanes, aliphatics, aldehydes, carboxylic acids, ketones, and esters. These natural active molecules found in *P. orientalis* bark can be used as drug and biomedical active ingredients for anti-cancer and anti-inflammatory purposes, demonstrating the bark extractives' wide-ranging potential as a raw material in various industries and agriculture.

*3.2. Effects of Catalysts on Pyrolysis Behavior of POL Bark*

3.2.1. TG Analysis

The effects of catalysts on the pyrolysis of *P. orientalis* bark have been investigated. Thermogravimetric analysis (TGA) results for *P. orientalis* bark (POL-B-$Co_3O_4$), POL-B-$Fe_2O_3$, and POL-B- $Co_3O_4$/$Fe_2O_3$ are shown in Figure 4. The primary components of *P. orientalis* bark include cellulose, lignin, hemicellulose, tannin, and extractives. Consequently, the actual behavior of the components during the thermal degradation of *P. orientalis* bark is quite complex due to the variable reactivity and stability of the different elements and the potential for interactions between them.

Figure 4 shows that the pyrolysis process included three distinct phases for each sample. At the beginning, when the temperature was raised from 30 to 200 °C, all of the samples lost weight due to water evaporation and the loss of volatile chemicals. The second stage of decomposition occurred between 200 and 450 °C [69]. During this stage, all DTG curves (Figure 4B1–B3) showed two clear weight-loss peaks due to cellulose, hemicellulose, and tannin decompositions [70,71]. According to a report by Yu[74], the extent of cellulose decomposition was small at 325 °C, with a high heating rate. However, higher temperatures enhanced cellulose decomposition, and complete conversion occurred at 450 °C, which is consistent with our results. The first peak can be attributed to the decomposition of hemicellulose. In contrast to cellulose and lignin, hemicellulose is the less stable component in *P. orientalis* bark and began to decompose at around 200 °C. This might be a result of low polymerization, such as that of cellulose and lignin [72]. The second peak can be attributed to the primary weight loss of cellulose and lignin in the temperature range of 320–450 °C.

Moreover, the main decomposition of tannin and ligin occurred between 200 and 450 °C. Due to its highly crosslinked nature, lignin has a lower reactivity than hemicellulose. Further, complex aromatic rings comprising different branched structures cause lignin to pyrolyze between 200 and 900 °C [21]. The third stage exhibited a moderate and relatively slow weight loss (490 °C). This could result from the primarily slow pyrolysis of lignin and the other solid residues of cellulose and hemicellulose.

From Figure 4A1,B1 and Table 5, it is clear that notable changes were found in the TG and DTG curves when nano-catalysts ($Co_3O_4$ and $Fe_2O_3$) were added. During the second stage (200–450 °C), two peaks were observed in the DTG curves of POL-B. With the addition of nano-$Fe_2O_3$, the temperature of peak 2 decreased from 306 °C (without a nano-catalyst) to 301 °C (with nano-$Fe_2O_3$), and the temperature of peak 3 decreased from 351 °C to 333 °C. The nano-$Fe_2O_3$ catalyst was shown to increase acid and ketone compound production during catalytic pyrolysis of cellulose and hemicellulose at lower temperatures. The inclusion of a nano-$Fe_2O_3$ catalyst significantly affected the pyrolysis of POL-B, as seen by the fluctuating peak and DTG values in the third stage. The weight loss for POL-B, POL-B-$Co_3O_4$, POL-B-$Fe_2O_3$, and POL-B-$Co_3O_4$/$Fe_2O_3$ at temperatures over 450 °C was 12.66, 11.98, 21.69 and 17.90%, respectively. The POL-B-$Fe_2O_3$ sample shows

the greatest reduction in mass when compared to the rest of the group. It showed that nano-$Fe_2O_3$ facilitated the last-stage pyrolysis of lignin and the residual solid residues of cellulose and hemicellulose.

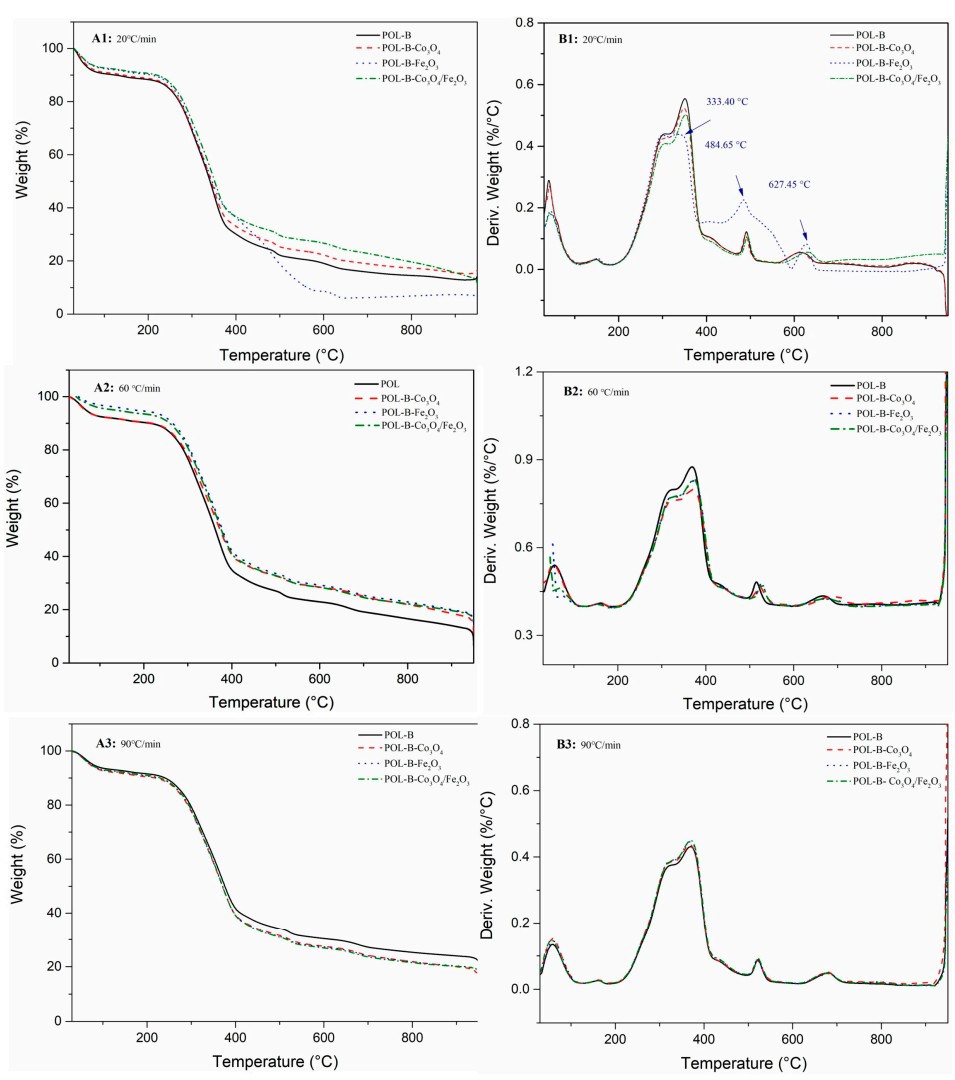

**Figure 4.** TG and DTG curves of POL-B, POL-B-$Co_3O_4$, POL-B-$Fe_2O_3$, and POL-B-$Co_3O_4$/Fe2O3 with different ratings: (**A1,B1**) 20 °C/min; (**A2,B2**) 60 °C/min; (**A3,B3**) 90 °C/min.

**Table 5.** Thermal properties of POL-B, POL-B-$Co_3O_4$, POL-B-$Fe_2O_3$, and POL-B-$Co_3O_4$/$Fe_2O_3$.

| Samples | Heating Rate (°C/min) | The First Stage: 30–200 °C | | The Second Stage: 200–450 °C | | | The Third Stage: 450–950 °C | | | Residues (%) |
|---|---|---|---|---|---|---|---|---|---|---|
| | | TP1 (°C) | Weight Loss (%) | $T_{P2}$ (°C) | $T_{P3}$ (°C) | Weight Loss (%) | $T_{P4}$ (°C) | $T_{P5}$ (°C) | Weight Loss (%) | |
| POL-B | 20 | 149 | 11.55 | 306 | 351 | 61.77 | 491 | 613 | 12.66 | 14.02 |
| | 60 | 161 | 9.66 | 324 | 369 | 60.35 | 515 | 666 | 16.86 | 13.13 |
| | 90 | 162 | 8.44 | 330 | 370 | 54.80 | 521 | 679 | 12.96 | 23.80 |
| POL-B -$Co_3O_4$ | 20 | 149 | 11.23 | 307 | 349 | 59.90 | 489 | 615 | 11.98 | 16.89 |
| | 60 | 161 | 9.57 | 326 | 368 | 55.05 | 523 | 691 | 17.49 | 17.89 |
| | 90 | 163 | 9.40 | 326 | 368 | 56.29 | 521 | 678 | 21.08 | 13.23 |
| POL-B -$Fe_2O_3$ | 20 | 153 | 9.75 | 302 | 333 | 61.55 | 485 | 627 | 21.69 | 7.01 |
| | 60 | 158 | 7.03 | 327 | 367 | 58.06 | 515 | 668 | 10.06 | 24.85 |
| | 90 | 163 | 9.13 | 331 | 369 | 56.91 | 520 | 680 | 13.85 | 20.11 |
| POL-B-$Co_3O_4$/$Fe_2O_3$ | 20 | 150 | 9.29 | 308 | 352 | 57.38 | 493 | 631 | 17.90 | 15.43 |
| | 60 | 160 | 8.49 | 338 | 373 | 57.09 | 528 | 664 | 15.43 | 18.99 |
| | 90 | 161 | 9.02 | 331 | 370 | 57.24 | 523 | 675 | 13.79 | 19.95 |

### 3.2.2. TG-FTIR Analysis

An FTIR spectrometer coupled with the thermogravimetric analyzer was used for the online detection of gas-phase products during the pyrolysis process. The three-dimensional FTIR spectrograms of the pyrolysis volatiles of *P. orientalis* bark, POL-B-Co$_3$O$_4$, POL-B-Fe$_2$O$_3$, and POL-B-Co$_3$O$_4$/Fe$_2$O$_3$, are shown in Figure 5. Six types of gases and four types of functional groups were investigated, and their IR profiles were presented in the FTIR spectra with time, following a linear relationship with the pyrolysis temperature. The TG-FTIR curves (Figure 5) show that in the first pyrolysis stage, when there was little weight loss, essentially no chemicals were detected. Then, FTIR was used to identify a wide variety of chemicals released during the pyrolysis process. Characteristic absorbances allowed for the identification of the common chemicals. H$_2$O was represented by absorption at roughly 3400 cm$^{-1}$, CH$_4$ at 3043–2637 cm$^{-1}$, CO$_2$ at 2380–2232 cm$^{-1}$, and CO at 2230–2072 cm$^{-1}$ [64]. It was determined that the stretching vibration of C=O was responsible for the absorption peaks at 1873 and 1618 cm$^{-1}$. This proved that the *P. orientalis* bark extracts included organic components such as aldehydes, ketones, and acids. The most prominent weight-loss peak in the DTG curve coincided with the 5–8 min period with the greatest intensity of pyrolysis volatiles.

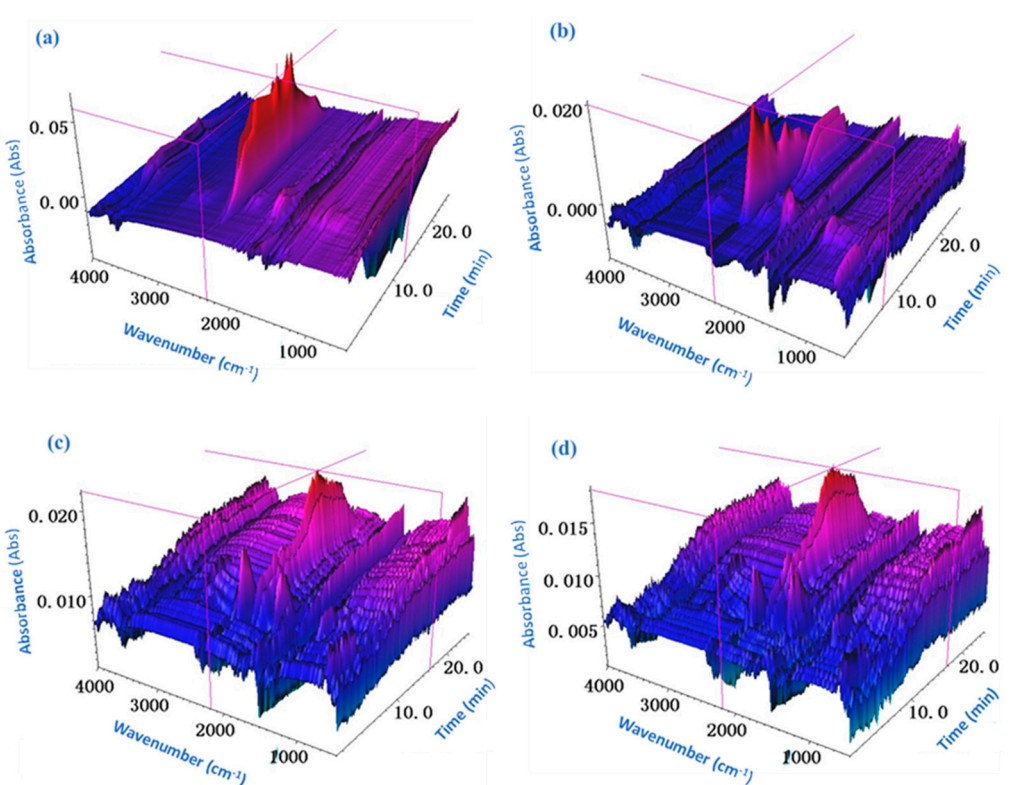

**Figure 5.** 3D FTIR spectrograms of pyrolysis volatiles of the samples: POL-B (**a**), POL-B-Co$_3$O$_4$ (**b**), POL-B-Fe$_2$O$_3$ (**c**), and POL-B-Co$_3$O$_4$/Fe$_2$O$_3$ (**d**).

### 3.2.3. Py-GC/MS Analysis

Under high temperatures, the pyrolysis of biomass is very complex. It involved multiple parallel and competing reactions, such as decomposition, dehydration, decarboxylation, decarbonylation, and depolymerization. In this study, the effects of the nano-Co$_3$O$_4$, Fe$_2$O$_3$, and Co$_3$O$_4$/Fe$_2$O$_3$ catalysts on the distribution and yield of pyrolysis products were investigated at 950 °C. Tables S2–S5 illustrate that the pyrolysis of *P. orientalis* bark changed in composition depending on the catalyst used. The chemical structure was used to categorize these compounds into groups for the easier presentation of the results: acids, aldehydes, ketones, esters, phenols, alkenes, alcohols, amines, olefins, furans, aromatics, and others. Likewise, in addition to the ways that biofuel may be used as petrol or diesel, it can also be

used as a chemical feedstock because of the high heating value of compounds like alkenes, olefins, and aromatics [72]. Table S2 shows the several-times-higher chemical content of *P. orientalis* bark.

The nano-catalysts provided strong evidence that the pyrolysis compositions were accurate, as seen in Figures 6 and 7. Aldehydes, ketones, olefins, acids, and esters were the most abundant compounds in both the catalytic and non-catalytic tests. The distribution of pyrolysis products was also affected by contrasting the effects of the catalyst type on these yields. According to the Py-GC-MS result of *P. orientalis bark* (Figure 7 and Table S2), 47 chemical compounds were identified in 67 peaks of *P. orientalis* bark. The high-content pyrolysis products included acetaldehyde (2.606%), furfural (0.453%), cyclobutylamine (0.206%), acetone (2.468%), acetic acid ethyl ester (1.418%), 2-Butene 4-Methyl-2-pentyne (0.169%), acetic acid (1.808%), toluene (0.219%), guaiacol (0.171%), furan (0.545%), and 2-methyl-Furan (0.512%). In the Py-GC-MS analysis of POL-B-$Co_3O_4$, more than 120 peaks were detected, and 67 chemical compounds were identified. In those of the POL-B-$Fe_2O_3$ and POL-B-$Co_3O_4$/$Fe_2O_3$ samples, 69 and 76 compounds were identified, respectively. Our data showed that nano-$Co_3O_4$ and nano-$Fe_2O_3$ distinctly increased the yield of phenols and aromatics compared to the non-catalytic sample. This phenomenon could result from the catalytic activity of nano-$Co_3O_4$ and nano-$Fe_2O_3$, which promoted lignin conversion to phenols, some of which were then further deoxygenated and converted to aromatics. A decreased ester yield was observed in nano-$Co_3O_4$ and nano-$Co_3O_4$/$Fe_2O_3$ catalytic experiments, most prominent in the presence of nano-$Co_3O_4$. A reduction in the amount of aldehyde could be found in catalytic nano-$Co_3O_4$ pyrolysis compared to nano-$Fe_2O_3$ pyrolysis, which promotes aldehyde yield. This result indicated that nano-$Fe_2O_3$ enabled a more balanced pyrolysis, thus producing more valuable chemical products.

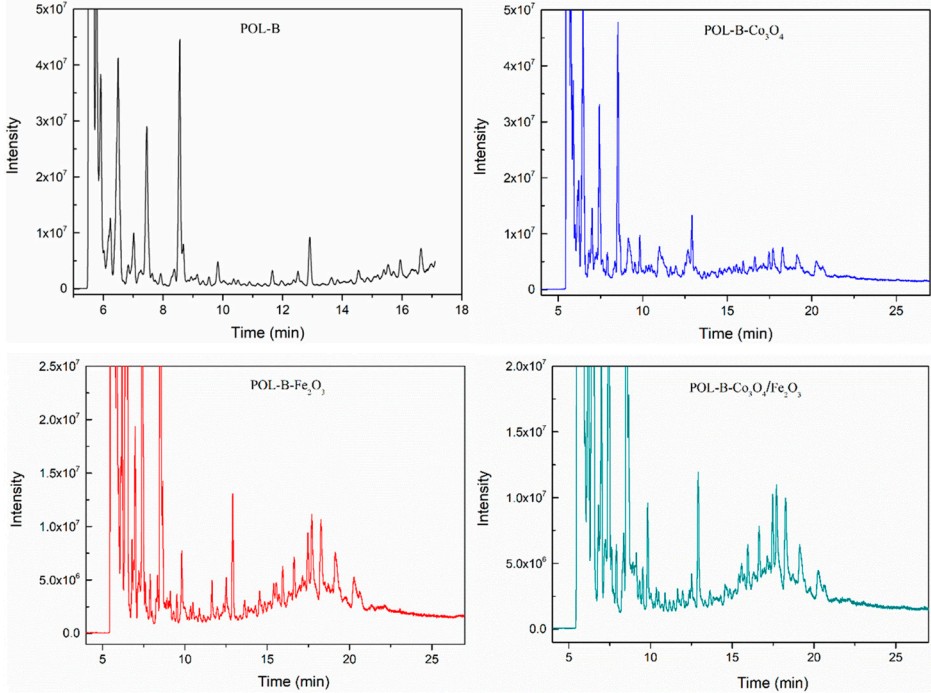

**Figure 6.** Typical ion chromatograms from non-catalytic and catalytic (with nano-$Co_3O_4$, $Fe_2O_3$, and $Co_3O_4$/$Fe_2O_3$) fast pyrolysis of *Platanus orientalis* Linn bark.

Acids and ketones produced in the presence of nano-catalysts (nano-$Co_3O_4$, nano-$Fe_2O_3$, and nano-$Co_3O_4$/$Fe_2O_3$) were observed to be higher than those obtained from the fast pyrolysis of POL-B without catalyst. This indicated that the nano-catalysts promoted the decomposition of hemicellulose in POL-B. In addition, the yield of olefins and alkanes increased under conditions of catalytic pyrolysis compared to non-catalytic *P. orientalis* bark

samples. It can be seen that the olefins produced in the presence of nano-$Fe_2O_3$ are higher than those derived from nano-$Co_3O_4$ and nano-$Co_3O_4/Fe_2O_3$.

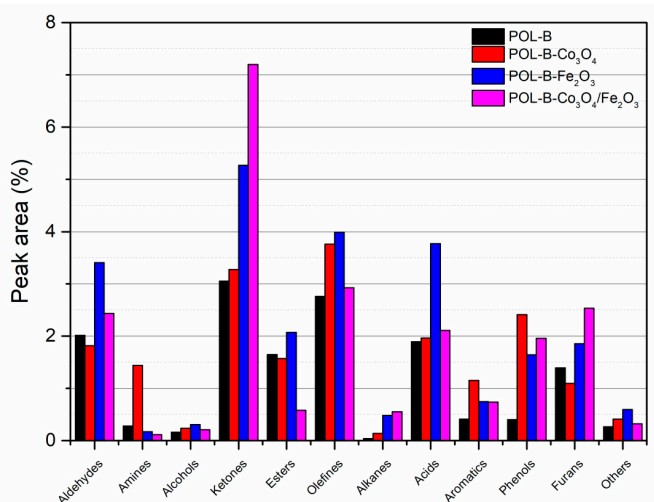

**Figure 7.** Different kinds of compounds from non-catalytic and catalytic (with nano-$Co_3O_4$, $Fe_2O_3$, and $Co_3O_4/Fe_2O_3$) fast pyrolysis of *Platanus orientalis* Linn bark.

Based on the above results, it can be found that the nano-catalysts can promote the pyrolysis of hemicellulose at a lower temperature, resulting in the production of more acids and ketone compounds. The nano-$Fe_2O_3$ catalyst favors the formation of furans, ketones, aromatics, aldehydes, alkanes, olefins and phenols, whereas the nano-$Co_3O_4$ catalyst promotes the formation of ketones, aromatics, olefins, and phenols.

## 4. Conclusions

In this study, *P. orientalis* bark was extracted and pyrolyzed to produce natural-product active molecules and high-quality bio-oil, and the products were analyzed using GC-MS, TOF-LC-MS, FTIR, NMR, and Py-GC-MS. GC-MS and results showed high concentrations and a wide variety of chemical compounds in the bark of *P. orientalis*, such as triacontanol, 1-Hexanol, Hexadecanoic acid, methyl ester, 2-ethyl-, β-Sitosterol, γ-Sitosterol, and 3,4,5-trimethoxy-Phenol. These extract compounds have promising future uses as raw materials in various industries and agriculture sectors due to the presence of natural-product active molecules that could be used as drugs and biomedically active ingredients in anti-inflammatory and anti-cancer drugs. In addition, methyl stearate, classified under ester compounds, can be used as a surfactant and lubricant. Hexadecanoic acid and methyl ester are only two examples of ester compounds that may be utilized as building blocks in emulsifiers, wetting agents, stabilizers, and plasticizers. The nano-catalysts promoted the production of aromatics, phenols, ketones, olefins, furans, and alkane compounds. According to the Py-GC-MS results, the catalyst type had a major impact on the pyrolysis of *P. orientalis* bark's components. For the Py-GC-MS data of POL-$Co_3O_4$, POL-$Fe_2O_3$, and PO-$Co_3O_4/Fe_2O_3$ samples, however, 67, 69, and 76 compounds were identified, respectively. The results showed that nano-$Co_3O_4$ and nano-$Fe_2O_3$ distinctly increase the yield of phenols and aromatics compared to non-catalytic samples. In addition, the nano-catalysts can promote the pyrolysis of hemicellulose at a lower temperature, resulting in the production of more acids and ketone compounds.

**Supplementary Materials:** The following supporting information can be downloaded at: https://www.mdpi.com/article/10.3390/f14102002/s1, Figure S1: Total ion chromatograms of *Platanus orientalis* Linn bark (extracted by methanol); Figure S2: Total ion chromatograms of *Platanus orientalis* Linn bark (extracted by ethanol/benzene, 1:1); Figure S3: Total ion chromatograms of *Platanus orientalis* Linn bark which was extracted by methanol/ethanol (1:1); Figure S4: 2D-HSQC NMR spectra of the extract of *Platanus orientalis* Linn bark by different solvent: methanol (E1),

ethanol/benzene (E2), methanol/ethanol (E3); Table S1: TOF-LC-MS analysis of the extractives of POL bark by methanol/ethanol; Table S2: Py-GC-MS analysis of POL-B; Table S3: Py-GC-MS analysis of POL-B-Co$_3$O$_4$; Table S4: Py-GC-MS analysis of POL-B-Fe$_2$O$_3$; Table S5: Py-GC-MS analysis of POL-B-Co$_3$O$_4$/Fe$_2$O$_3$.

**Author Contributions:** Conceptualization, H.L. and Q.Z.; methodology, C.L. and H.L.; software, Y.Z. and Y.G.; validation, Q.Z. and Y.Z.; formal analysis, H.L., Y.Z., J.L., Z.Z., N.Z. and C.L.; investigation, H.L., Y.Z., Z.Z. and C.L.; resources, H.L. and C.L.; data curation, Y.Z., J.L., Z.Z., Y.G. and R.Y.; writing—original draft, H.L., Y.Z., Q.Z. and C.L.; writing—review and editing, H.L., Q.Z. and C.L.; supervision, Q.Z. and C.L.; project administration, H.L. and C.L.; funding acquisition, H.L. and C.L. All authors have read and agreed to the published version of the manuscript.

**Funding:** This work was financially supported by the Special Fund for Young Talents in Henan Agricultural University (30500928), the Key scientific research projects of institutions of high education in Henan (21A220001), and the Key R&D and Promotion Special Project of Henan Province (232102230108 and 232102320346).

**Data Availability Statement:** Not applicable.

**Conflicts of Interest:** The authors declare no conflict of interest.

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
