# Peer review of "The Potential of Platanus orientalis L. Bark for High-Grade Resource Utilization"

_forests, doi:10.3390/f14102002_

Round 1

Reviewer 1 Report

Comments on the article “The high-grade resource utilization of Platanus orientalis bark: extraction compounds and the pyrolysis productions with nano catalystics”.

Abstract

The abstract presents some information in a run-over form. The objective is presented even before the species is identified.

“In this study, the solvent extracts were analyzed…”, what kind of solvents? From where? The text needs to be clearer.

Nano-catalysts or nano catalysts? Standardize across the text

Introduction

The study focuses on the title, the issue of nano catalysts, but this is little focused on in the introduction. Nano catalysts should be more present in the introduction text, citing their advantages and possibilities in relation to other existing ones, after all, this is what the work contributes to science.

Any information about the wood of the mentioned species for energy use?

Materials and Methods

Line 117 - Use the units in the International System (hours = h)

Photos of the extraction process of solvents from the bark of the species in schematic format.

Line 121 – 150: Units such as mm, mg, and µm must be separated from digits.

Results and discussion

This section needs to be reworked. Discussions should accompany the graphs or tables in which they are presented. Authors should not present a range of graphs and then present a lengthy discussion mixing all the points. In addition, the figure should be called out in the text before presenting the figure, and not start the results by randomly dropping a figure. The author should be more careful when presenting the results.

Conclusion

The conclusion is inconclusive and unproductive. The author does not present direct points that allow classifying the bark as a differential for chemical extraction. The author should address the elements that stand out and mention to what use it would be forwarded.

Few errors, but the understanding of the information is not compromised

Author Response

Thank you for your letter with the reviewers’ valuable comments and suggestions concerning our manuscript entitled “The potential of Platanus orientalis L. bark for high-grade resource utilization” (forests-2558917). We appreciate the opportunity to modify our manuscript according to the valuable comments and suggestions from the editors and reviewers.

Careful corrections have been made to the manuscript based on the received comments. In the revised version, all changes were marked in red. We hope the revised manuscript will meet your approval. Please find the summary of our detailed replies to the editors’ and reviewers’ comments/questions below.

Reviewer 2 Report

1.       The title is confusing and needs to be rewritten around the objective of the paper.

2.       The grammar in some places is not that good.

3.       Was there a standard associated with the extraction method?

4.       Not all of the extractive compounds in bark were identified in the literature review and can be found in Forest Prod. J. 73(3):194–208. For example, this reference shows that triterpenes like lupeol, botulin, and betulinic acid can be extracted from bark to achieve anticancer properties.  It would be nice to be more specific about that compounds provide functional benefits.

5.       The conclusions section mostly speculates about the potential benefits of this work and this is not really a conclusion.  The conclusions section does not well summarize the work that was done.

Minor revision

Author Response

Dear editor and Reviewers:

Thank you for your letter with the reviewers’ valuable comments and suggestions concerning our manuscript entitled “The potential of Platanus orientalis L. bark for high-grade resource utilization” (forests-2558917). We appreciate the opportunity to modify our manuscript according to the valuable comments and suggestions from the editors and reviewers.

Careful corrections have been made to the manuscript based on the received comments. In the revised version, all changes were marked in red. We hope the revised manuscript will meet your approval. Please find the summary of our detailed replies to the editors’ and reviewers’ comments/questions below.

Round 2

Reviewer 1 Report

No more comments, the manuscript can be accepted for publication.